# Cytotoxic Xanthones from *Hypericum stellatum*, an Ethnomedicine in Southwest China

**DOI:** 10.3390/molecules24193568

**Published:** 2019-10-02

**Authors:** Yuanyuan Ji, Ruifei Zhang, Chen Zhang, Xingyu Li, Adam Negrin, Chaonan Yuan, Edward J. Kennelly, Chunlin Long

**Affiliations:** 1College of Life and Environmental Sciences, Minzu University of China, Beijing 100081, China; 18363591863@163.com (Y.J.); ruifeizhang@yeah.net (R.Z.); zcyx3120@163.com (C.Z.); 13161131400@163.com (C.Y.); 2Key Laboratory of Ethnomedicine, Minzu University of China, Ministry of Education, Beijing 100081, China; 3College of Science, Yunnan Agriculture University, Kunming 650201, China; lixingyu@ynau.edu.cn; 4Department of Biological Sciences, Lehman College, City University of New York, Bronx, NY 10468, USA; adamnegrin@gmail.com (A.N.); edward.kennelly@lehman.cuny.edu (E.J.K.); 5The Graduate Center, City University of New York, New York, NY 10016, USA

**Keywords:** *Hypericum stellatum*, xanthone, liver carcinoma cell lines, cytotoxicity, ethnomedicine

## Abstract

*Hypericum stellatum*, a species endemic to China, is used to treat hepatitis by several ethnic groups in Guizhou Province. This research was inspired by the traditional medicinal usage of *H. stellatum*, and aims to explore the phytochemistry and bioactivity of *H. stellatum* to explain why local people in Guizhou widely apply *H. stellatum* for liver protection. In this study, two new prenylated xanthones, hypxanthones A (**8**) and B (**9**), together with seven known compounds, were isolated from the aerial parts of the plant. Spectroscopic data as well as experimental and calculated ECD spectra were used to establish the structures of these compounds. Six xanthones isolated in this study, together with four xanthones previously isolated from *H. stellatum*, were evaluated for their growth-inhibitory activities against five human liver carcinoma cell lines to analyze the bioactivity and structure-activity relationship of xanthones from *H. stellatum*. Isojacareubin (**6**) showed significant cytotoxicity against five human liver carcinoma cell lines, with an IC_50_ value ranging from 1.41 to 11.83 μM, which was stronger than the positive control cisplatin (IC_50_ = 4.47–20.62 μM). Hypxanthone B (**9**) showed moderate cytotoxicity to three of the five cell lines. Finally, structure-activity analysis revealed that the prenyl and pyrano substituent groups of these xanthones contributed to their cytotoxicity.

## 1. Introduction

The plants in the genus *Hypericum* have a long history of use as herbal medicines in China. There are 64 *Hypericum* species (33 endemic) distributed throughout China [1], and 19 of these species have been used traditionally as medicinal plants with a range of purported benefits, including antidepressant, detoxification, hemostasis, antibacterial, and hepatoprotection activities. Based on ethnobotanical investigations in southeast Guizhou province, eight *Hypericum* species were found to be used as ethnomedicines by several ethnic groups, including the Shui, Miao, Dong, and She peoples. Previous phytochemical investigations on *Hypericum* have reported various types of compounds, including naphthodianthrones, flavonoids, prenylated phloroglucinols, and phenols [2,3,4,5]. Furthermore, *Hypericum* extracts have been found to exhibit various pharmacological activities, including antidepressant, antiviral, antibacterial, and anti-HIV activities [6,7,8,9,10].

*Hypericum stellatum* is an endemic species to China, primarily distributed in Chongqing and surrounding regions. Our ethnobotanical investigations in Majiang, a county of the Qiandongnan Miao and Dong Autonomous Prefectures in Guizhou Province, have shown that *H. stellatum* is used by the local Miao and Buyi people to treat liver diseases such as icterohepatitis. However, no chemical studies had been conducted on this species. In our recent research on *H. stellatum*, UPLC-Q-TOF-MS was applied to analyze the chemical constituents of the plant. The results showed that *H. stellatum* was rich in flavonoids, phenolic acids, and polycyclic polyprenylated acylphloroglucinols. *H. stellatum* possesses strong antioxidant activity and a high content of total phenols. Ten compounds, including flavonoids, xanthones, and phenolic acids, were isolated [11]. To further illuminate the plant′s pharmacological constituents, we used column chromatography (CC), including silica gel, Sephadex LH-20, and recycling preparative HPLC, which yielded two new xanthones and seven known compounds. Six xanthones (**1**, **2**, and **6**–**9**) obtained in this research, together with four previously isolated xanthones (**3**–**5** and **10**), were tested for cytotoxicity against five human liver carcinoma cell lines. Herein, we report the isolation, structure elucidation, and bioactivities of these compounds isolated from *H. stellatum*.

## 2. Results and Discussion

A 95% aq. EtOH extract of the aerial part of *H. stellatum* was suspended in water and extracted successively with petroleum ether, EtOAc, and *n*-BuOH. The EtOAc extract showed strong antioxidant activity and a high total phenol content. The fraction of EtOAc was repeatedly subjected to silica gel, Sephadex LH-20, and a liquid chromatography loop preparation, yielding two new prenylated xanthones (**8** and **9**) and seven known compounds. Structures of the new compounds were elucidated using spectroscopic data and calculated via ECD spectra, the spectroscopic data can be seen in Appendix A.

Hypxanthone A (**8**) was obtained as a yellow amorphous powder. Its molecular formula, C_18_H_16_O_7_, was established by HR-ESI-MS (*m*/*z* 343.0826 [M − H]^−^, calcd. 343.0823), with 11 degrees of unsaturation. The ^1^H-NMR data (Table 1) exhibited signals of one methyl group (δ_H_ 1.92 (3H, s)), four aromatic protons (δ_H_ 5.63 (1H, t, *J* = 7.2 Hz), 6.22 (1H, s), 6.89 (1H, d, *J* = 8.4 Hz), 7.85 (1H, d, *J* = 8.4 Hz)), and two methylene groups (δ_H_ 3.63 (2H, d, *J* = 7.2 Hz), 3.91 (2H, s)). The ^13^C-NMR data of compound **8** revealed the presence of 18 carbons, including one carbonyl, six oxygenated sp^2^ tertiary carbons, four sp^2^ quaternary carbons, four aromatic methines, two methylenes, and one methyl group. These results indicated that **8** has a skeleton of prenylated xanthone. Besides, the presence of a prenyl group in **8** was deduced from signals at δ_H_ 3.63/δ_C_ 22.0 (C-1′), δ_H_ 5.63/δ_C_ 125.0 (C-2′), δ_H_ 3.91/δ_C_ 69.1 (C-4′), and δ_H_ 1.92/δ_C_ 14.0 (C-5′). The prenyl group was assigned to locate at C-4 based on the HMBC correlations of H-1′/C-3, H-1′/C-4, H-1′/C-10a, H-1′/C-3′, H_3_-5′/C-2′, and H_3_-5′/C-4′, as shown in Figure 1. The ROESY correlation of H-2′/H-4′ suggested that the Δ^2′(3′)^ double bond was *E* configured, thus defining the structure (Figure 1), and this was named hypxanthone A.

Hypxanthone B (**9**) was obtained as yellow amorphous powder. Its molecular formula, C_23_H_22_O_6_, was established by HR-ESI-MS (*m*/*z* 393.1341 [M − H]^−^, calcd. 393.1344), with 13 degrees of unsaturation. The ^1^H-NMR data (Table 1) showed resonances characteristic for three methyl groups (δ_H_ 1.79 (3H, s), 1.80 (3H, s), 1.80 (3H, s)) and five aromatic protons (δ_H_ 4.92 (2H, s), 5.28 (1H, d, *J* = 9.0 Hz), 6.14 (1H, s), 6.90 (1H, d, *J* = 8.4 Hz), 7.60 (1H, d, *J* = 8.4 Hz)). The ^13^C-NMR data were similar to those of **8**, except for the additional signals of two methyls, two aromatic carbons, and one methine carbon. The 1D-NMR data indicated that **9** also possesses a prenylated xanthone skeleton. The HSQC and DEPT spectra suggested that there were two double bonds (one terminal double bond), three methyl groups, one methylene, and two methines (one oxygenated carbon). Considering the molecular formula unsaturation, there should be another ring in **9**. The ^1^H–^1^H COSY spectrum displayed the correlations of H-1′/H-2′/H-3′/H-4′. The HMBC spectrum showed cross peaks from H-1′ to C-5/C-6/C-10a, H-3′ to C-5′/C-8′, H-7′ to C-4′/C-5′, and H-9′ to C-2′/C-10′, as shown in Figure 2, thus assigning the planar structure. The (2′β, 3′α) orientations of substituents in **9** were deduced from the ROESY correlations between H-1′a and H-2′, H-1′b, and H-3′, as shown in Figure 2. The (2′*S*, 3′*S*) absolute configuration was established via comparison of the experimental and calculated ECD spectra, which displayed cotton effects at 219, 249, and 310 nm (Figure 3). Thus, the structure of hypxanthone B was elucidated as depicted below.

In addition to hypxanthones A (**8**) and B (**9**), seven known compounds were isolated and identified by comparison with their spectroscopic data in the literatures and standard compounds. These known compounds were the following: 2-hydroxy-3-methoxyxanthone (**1**) [12], 1,3,8-trihydroxyxanthone (**2**) [13], isojacareubin (**6**) [14], 1,3,7-trihydroxy-6-methoxyxanthone (**7**) [15], ethyl 2-[(3,4-dihydroxybenzoyloxy)-4,6-dihydroxyphenyl] acetate [16], β-sitosterol, and daucosterol.

To evaluate the bioactivity and structure-activity relationship of xanthones from *H. stellatum*, the xanthones isolated in this research, together with five previously isolated xanthones, 1,3,6,7-tetrahydroxylxanthone (**3**), 1,3,7-trihydroxyxanthone (**4**), 3,6,7-trihydroxy-1-methoxyxanthone (**5**) and calycinoxanthon D (**10**), were tested for their growth-inhibitory activities against five human liver carcinoma cell lines (SMMC-7721, Huh-7, HepG2, SK-HEP-1, PLC/PRF/5) and against the human normal liver cell line LO2. The structures of the xanthones are shown in Figure 4.

Cytotoxic activity tests (Table 2) showed that simply oxygenated xanthone compounds **3** and **5** showed weak cytotoxicity against all five cell lines with IC_50_ values > 40 µM. This is probably due to the absence of the groups which are crucial to locking the compound into the domain in the target binding site. Compounds **6**, **9** (hypxanthone B), and **10** showed strong cytotoxic activity towards SMMC-7721 cells, with IC_50_ values ranging from 1.41 to 8.26 μM, while **2**, **7**, and **8** (hypxanthone A) showed moderate cytotoxic activity against the same cell line. Compound **6** (IC_50_ = 9.09 µM) showed strong cytotoxic activity towards Huh-7 cells, and compounds **9** and **10** showed moderate cytotoxic activity towards Huh-7 cells. Compounds **1**, **4**, **9**, and **10** showed moderate cytotoxic activity towards HepG2 cells, and **6** showed strong cytotoxic activity with an IC_50_ value of 2.40 µM. Tests on SK-HEP-1 cells showed that compounds **7** and **10** showed moderate cytotoxic activity, and **6** showed strong cytotoxic activity (IC_50_ = 9.20 µM). Compounds **6**, **9**, and **10** showed moderate cytotoxic activity towards PLC/PRF/5 cells.

Compounds which showed strong cytotoxic activity were shown to have prenyl groups, including compounds **9** and **10**. Isojacareubin (**6**), with its pyrano substituent group, showed strong cytotoxic activity to four human liver carcinoma cell lines (SMMC-7721, Huh-7, HepG 2, SK-HEP-1), and its cytotoxic activity was greater than that of the positive control cisplatin (Meilunbio, CAS No. 15663-27-1). However, it also exhibited strong cytotoxic activity to human normal liver cell line LO2. Isojacareubin has been isolated from many *Hypericum* species, including *Hypericum sarothranol* [17], *Hypericum roeperanum* [18], *Hypericum japonicum* [19] and *Hypericum henryi* [20], and it has been found to be the most potent cytotoxic compound against various cancer cells including HeLa, A549, PANC-1, HL-7702 [21], AGs, MCF7, MDAMB-231, and U87 [22]. Considering their chemical structure-activity relationships, the prenyl and pyrano substituent groups of the xanthone derivatives likely contributed to the cytotoxicity of these compounds against human liver carcinoma cell lines (SMMC-7721, Huh-7, HepG2, SK-HEP-1, PLC/PRF/5).

As active secondary metabolites, xanthones commonly occur in various herbal medicines; a total of 168 species of herbal plants belonging to 58 genera and 24 families have been found to contain xanthones. Calophyllaceae, Gentianaceae and Clusiaceae (Guttiferae) are the most widely distributed families containing xanthones [23]. In Clusiaceae sensu lato, two genera (*Hypericum* and *Garcinia)* are rich in variously oxidized and prenylated xanthones, such as *Hypericum uralum* [24], *Hypericum monogynum* [25], *Hypericum riparium* [26], *Garcinia nujiangensis* [22], *Garcinia cowa* [21], and *Garcinia mangostana* [27]. Many of the compounds isolated from these plants could have remarkable medicinal potential, such as the prenylated xanthone gambogic acid isolated from *Garcinia hanburyi*. The resin of *G. hanburyi*, called *gamboge* in traditional Chinese medicine and ethnomedicine in Asian countries, possesses broad-spectrum anticancer activity and showed safety in Chinese phase II clinical trials carried out in 2009 [28]. Furthermore, our own research, as well as previous studies, has revealed that isojacareubin possesses potential medicinal value. Isojacareubin has been synthesized, and was found to be a potent inhibitor of PKC; these findings identify isojacareubin as a promising lead compound for the development of new antihepatoma agents [29].

Our previous study revealed that the antioxidant ability and total phenol content of EtOAc and *n*-BuOH extract showed little differences. However, the metabolite profiles of two extractions metabolic profile showed obvious differences. As a result, the *n*-BuOH extract should likely contain different biomarkers [11]. Further phytochemical research on the *n*-BuOH extraction will be carried out to search for candidates lead compounds.

## 3. Materials and Methods

### 3.1. General Experimental Procedures

One-dimensional (1D) and two-dimensional (2D) nuclear magnetic resonance (NMR) spectra were recorded on Bruker DRX-600 NMR spectrometers (Bruker, Bremerhaven, Germany), with tetramethylsilane (TMS) as an internal standard. Mass spectrometry (MS) data were collected using Shimadzu liquid chromatography-mass spectrometry-ion-trap-time of flight (LCMS-IT-TOF) (Shimadzu, Kyoto, Japan). Column chromatography was performed using silica gel (100–200 mesh and 200–300 mesh, Qingdao Haiyang Chemical, Inc. Qingdao, China), Sephadex LH-20 (GE Healthcare Bio-Sciences AB, Uppsala, Sweden; 2.5 cm × 150 cm, amount of resin 160 g, at a flow rate of 1 mL/min), and precoated TLC with silica gel 60 GF254 (Qingdao Haiyang Chemical, Inc.). The recycling preparative HPLC system LC-908-G30 (JAI, Japan Analytical Industry Co., Ltd., Tokyo, Japan), equipped with a JAIGEL-ODS-AP-L, SP-120-15 column (Serial No. 051,205,128 Japan Analytical Industry Co., Ltd., Tokyo, Japan) was used in sample purification.

### 3.2. Plant Material

The aerial parts of *Hypericum stellatum* N. Robson were collected from Majiang County, Qiandongnan Miao and Dong Autonomous Prefectures, Guizhou Province, China, in July 2016. The plant was identified by Jun Yang, a taxonomist at the Kunming Institute of Botany, Chinese Academy of Sciences. A voucher specimen (No. LongCL-060) has been deposited at the Key Laboratory of Economic Plants and Biotechnology, Kunming Institute of Botany.

### 3.3. Extraction and Isolation

The powdered, air-dried aerial parts of *H. stellatum* (5 kg) were extracted with 95% aq. EtOH (3 × 16 L, 3 h) and subsequently concentrated under vacuum to yield a crude extract (920.0 g). The crude extract was suspended in H_2_O (4 L) and successively extracted with petroleum ether (4.0 L × 3), EtOAc (4.0 L × 3) and *n*-BuOH (4.0 L × 3), to obtain the petroleum ether extract (211.0 g), EtOAc extract (390.0 g), and *n*-BuOH extract (295.0 g). The EtOAc extract (390.0 g) was subjected to silica gel CC, and eluted with CH_2_Cl_2_/CH_3_OH (50:1 to 0:1, *v*/*v*), yielding 11 fractions (Fr. 1–Fr. 11). Fr. 3 was subjected to silica gel CC and eluted with CHCl_3_/CH_3_OH (100:1 to 1:1, *v*/*v*) to afford six fractions (Fr. A1–Fr. A6). Fr. A2 fractionated by Sephadex LH-20 (CHCl_3_/CH_3_OH 1:1, *v*/*v*) into four fractions (Fr. A2a–Fr. A2d). Fr. A2a was submitted to recycling preparative HPLC (CH_3_OH) to afford 2-hydroxy-3-methoxyxanthone (4.5 mg) and 1,3,8-trihydroxyxanthone (3.9 mg). Fr. 5 was subjected to silica gel CC using (50:1 to 4:1, *v*/*v*) to afford 10 fractions (Fr. B1–Fr. B10). Fr. B4 was purified by Sephadex LH-20 (CHCl_3_/CH_3_OH 1:1, *v*/*v*) and submitted to recycling preparative HPLC (CH_3_OH) to afford isojacareubin (1.5 mg). Fr. 7 was subjected to silica gel CC using (40:1 to 10:1, *v*/*v*) to afford eight fractions (Fr. C1–Fr. C8). Crystallization of Fr. C4 yielded 1,3,7-trihydroxy-6-methoxyxanthone (1.8 mg). Fr. C5 was fractionated by Sephadex LH-20 (CHCl_3_/CH_3_OH 1:1, *v*/*v*) into five fractions (Fr. C5a–Fr. C5e). Fr. C5a was submitted to recycling preparative HPLC (CH_3_OH) to afford ethyl 2-[(3,4-dihydroxybenzoyloxy)-4,6-dihydroxyphenyl] acetate (1.7 mg) and β-sitosterol (3.5 mg). Fr. 8 was subjected to silica gel CC using (50:1 to 4:1, *v*/*v*) to afford 10 fractions (Fr. D1–Fr. D10). Fr. D7 was fractionated by Sephadex LH-20 (CHCl_3_/CH_3_OH 1:1, *v*/*v*) into three fractions (Fr. D7a–Fr. D7c). Fr. D7b was submitted to recycling preparative HPLC (CH_3_OH) to afford hypxanthone A (1.4 mg), hypxanthone B (1.2 mg), and daucosterol (4.5 mg).

### 3.4. ECD Calculations

The method used for ECD calculation has been previously reported [30]. The ROESY spectra were used to initially determine the relative configuration of hypxanthone B, followed by the MMFF94s force field (random conformational analysis). The B3LYP/6-31G(d) level of time-dependent density functional theory (TDDFT) was used to optimize the obtained conformers and was followed by ECD calculations via the TDDFT method (B3LYP/6-31+G(d), CPCM model = MeOH), with SpecDis v1.51 (with a half-bandwidth of 0.3 eV) to simulate the Boltzmann-weighted ECD spectra. The Gaussian 09 electronic structure package (version 7.0, Gaussian, Inc., Wallingford, CT, USA) was used to perform all of the calculations.

### 3.5. Characterization

Hypxanthone A. Yellow amorphous powder; UV (MeOH) λmax (log ε) 327 (4.32) nm; ^1^H- and ^13^C-NMR data: see Table 1; HRESIMS *m*/*z* 343.0826 [M − H]^−^ (calcd. for C_18_H_15_O_7_, 343.0823).

Hypxanthone B. Yellow amorphous powder; $[\alpha {]}_{D}^{25}$ +75 (c 0.04, MeOH); UV (MeOH) λmax (log ε) 330 (3.44) nm; ECD (c 1.01 × 10^−3^ M, MeOH) λmax (∆ε) 210 (+33.3), 250 (+18.4), 326 (+3.5) nm; ^1^H- and ^13^C-NMR data: see Table 1; HRESIMS *m*/*z* 393.1341 [M − H]^–^ (calcd. for C_23_H_19_O_6_, 393.1344).

### 3.6. MTS Assay

Considering its purported medicinal value for the treatment of liver disease, we tested the cytotoxic activities of ten xanthones isolated from *H. stellatum* against five human liver carcinoma cell lines (SMMC-7721, Huh-7, HepG2, SK-HEP-1, PLC/PRF/5), as well as the immortalized non-cancerous human liver cell (LO2) by the MTS method, in vitro, with cisplatin (CAS No. 15663-27-1, Meilunbio, Dalian, China) and Taxol (CAS No. 33069-62-4, Meilunbio, Dalian, China) as positive controls.

The isolated compounds were tested in vitro for their cytotoxicity to five human liver carcinoma cell lines (SMMC-7721, Huh-7, HepG2, SK-HEP-1, PLC/PRF/5) and the immortalized noncancerous human liver cell (LO2), in a 3-(4,5-dimethylthiazol-2-yl)-5(3-carboxymethoxyphenyl)2-(4-sulfopheny)-2*H*-tetrazolium (MTS; Promega, Beijing, China) assay [31]. In general, cells in the log phase of their cycle were seeded in 96-well plates (4000–5000 cells/well, NEST Biotechnology, Wuxi, China) in a 100-μL volume. After 12 h of incubation at 37 °C, each test compound was added. The cancer cell lines were exposed to the test compounds at five concentrations (0.064, 0.32, 1.6, 8, and 40 μM) in triplicate with cisplatin (Meilunbio, CAS No. 15663-27-1) and Taxol (Meilunbio, CAS No. 33069-62-4) as the positive control. After incubation for 48 h at 37 °C, 20 μL of MTS solution and 100 μL DMEM were added into the well. The incubation continued for another 1–4 h. The absorbance was measured at the detection wavelength of 490 nm (*L*1) and the reference wavelength of 680 nm (*L*2), and cytotoxicity for each compound was expressed as IC_50_ values by Reed and Muench′s method [32].

## 4. Conclusions

In summary, two new compounds, hypxanthone A (**8**) and hypxanthone B (**9**), and seven known compounds (2-hydroxy-3-methoxyxanthone (**1**), 1,3,8-trihydroxyxanthone, isojacareubin (**2**), Isojacareubin (**6**) 1,3,7-trihydroxy-6-methoxyxanthone (**7**), ethyl 2-[(3,4-dihydroxybenzoyloxy)-4,6-dihydroxyphenyl] acetate, β-sitosterol, and daucosterol) were isolated and identified from *H. stellatum*. Ten xanthones (six xanthones including compounds **1**, **2**, and **6**–**9** isolated in this research, along with four xanthones including **3**–**5** and **10** isolated previously) isolated from *H. stellatum* were tested for their cytotoxicity against five human liver carcinoma cell lines. Isojacareubin showed stronger cytotoxic activity than the positive control cisplatin; however, it was weaker than Taxol. Analysis of the structure-activity relationship suggested that xanthone derivatives bearing substituent groups such as prenyl and pyrano groups can improve cytotoxicity towards the tested cancer cell lines. These results indicated that *H. stellatum* is rich in xanthones. The indigenous knowledge leading local people in Guizhou to apply *H. stellatum* for liver protection and to treat hepatitis needs further research to reveal in order to reveal the mechanisms.

## Figures and Tables

**Figure 1 molecules-24-03568-f001:**
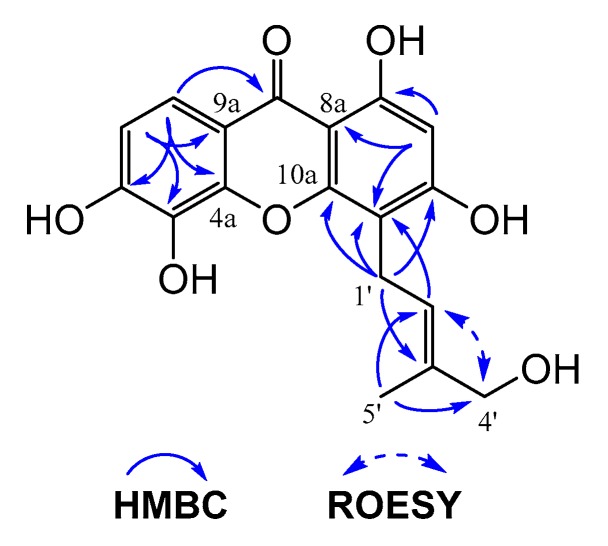
Key HMBC and ROESY correlations of hypxanthone A.

**Figure 2 molecules-24-03568-f002:**
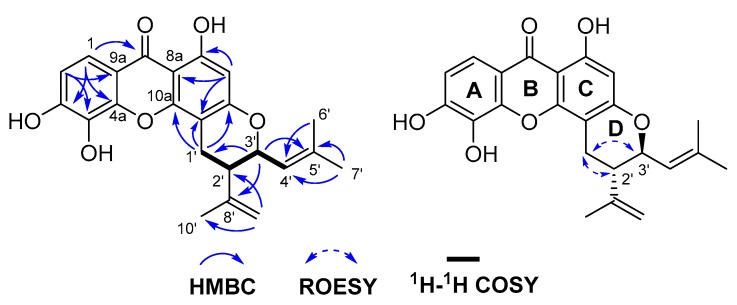
Key ^1^H–^1^H COSY, HMBC and ROESY correlations of hypxanthone B.

**Figure 3 molecules-24-03568-f003:**
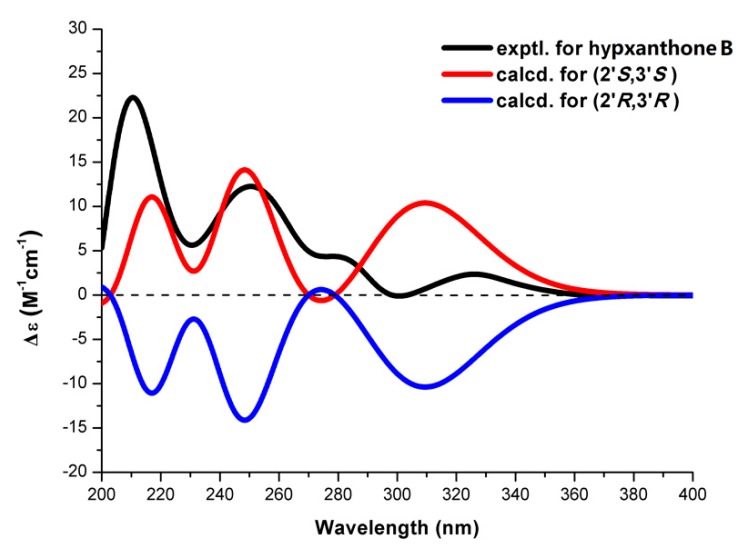
Comparison of experimental (black line) and calculated (red and blue lines) ECD spectra of hypxanthone B.

**Figure 4 molecules-24-03568-f004:**
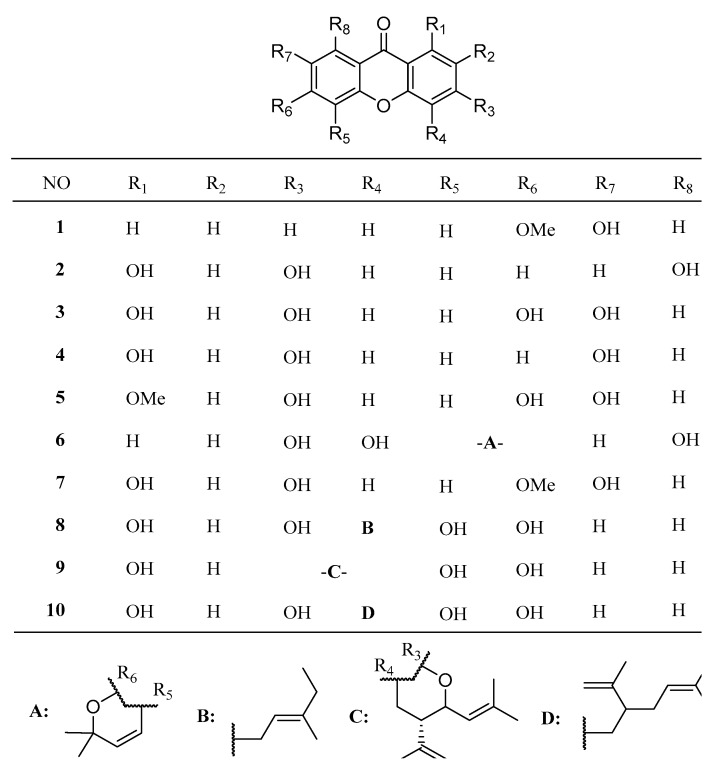
Structures of xanthone derivatives from *H. stellatum*.

**Table 1 molecules-24-03568-t001:** ^13^C- and ^1^H-NMR spectroscopic data (151/600 MHz) for hypxanthone A and hypxanthone B in methanol-*d*_4_ (δ in ppm, *J* in Hz).

Position	Hypxanthone A		Hypxanthone B	
	δ_C_	δ_H_	δ_C_	δ_H_
1	162.6		161.9	
2	98.3	6.22, s	99.2	6.14, s
3	164.4		162.8	
4	107.7		102.6	
5	133.4		133.9	
6	148.4		148.0	
7	113.4	6.89, d, 8.4	113.8	6.90, d, 8.4
8	117.5	7.85, d, 8.4	117.7	7.60, d, 8.4
9	182.4		182.1	
4a	153.2		153.5	
8a	103.2		103.8	
9a	114.8		115.0	
10a	156.3		156.0	
1′	22.0	3.63, s	25.4	2.90, dd (11.4, 16.2)
2′	125.0	5.63, t, 6.0	45.7	2.50, m
3′	135.9		77.3	4.80, t, 9.0
4′	69.1	3.91, s	124.2	5.27, d, 9.0
5′	14.0	1.92, s	140.2	
6′			25.9	1.80, s
7′			18.6	1.80, s
8′			146.4	
9′			113.7	4.92, s
10′			20.6	1.80, s

**Table 2 molecules-24-03568-t002:** IC_50_ values of xanthones from *H. stellatum.*

Compds	IC_50_ ± SD (μM)					
	SMMC-7721	Huh-7	HepG2	SK-HEP-1	PLC/PRF/5	LO2
**1**	>40	>40	10.19 ± 0.12	>40	>40	14.47 ± 0.95
**2**	15.20 ± 0.27	>40	>40	>40	>40	>40
**4**	>40	>40	22.60 ± 1.43	>40	>40	>40
**6**	1.41 ± 0.03	9.09 ± 0.38	2.40 ± 0.02	9.20 ± 0.21	11.83 ± 0.56	2.03 ± 0.04
**7**	28.18 ± 0.89	>40	>40	37.09 ± 0.97	>40	12.09 ± 0.14
**8**	27.56 ± 0.68	>40	>40	>40	>40	>40
**9**	8.26 ± 0.57	25.77 ± 2.04	11.93 ± 0.10	>40	30.76 ± 0.38	>40
**10**	6.27 ± 0.16	16.65 ± 0.24	21.33 ± 0.16	31.11 ± 2.67	24.89 ± 0.46	12.21 ± 0.25
Cisplatin *^a^*	4.47 ± 0.27	16.00 ± 0.95	10.29 ± 0.50	20.62 ± 1.03	10.66 ± 0.80	13.93 ±0.87
Taxol *^a^*	0.18 ± 0.03	0.11 ± 0.011	<0.01	<0.01	<0.01	<0.01

IC_50_ ≤ 10 = strong activity, 10 < IC_50_ ≤ 40 = moderate activity, IC_50_ > 40 = weak activity. Each date represents the mean of three independent experiments. ^a^ Positive control.

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
