# Peer review of "Cytotoxic Xanthones from Hypericum stellatum, an Ethnomedicine in Southwest China"

_molecules, 2019, doi:10.3390/molecules24193568_

Round 1
Reviewer 1 Report
The structure elucidation was clearly presented and well discussed. The English language could be improved a little more. There some correction that authors should make in text that are given in the attached file.

Author Response
Thank you very much!
Our co-authors, Professor Edward Kennelly and Dr. Adam Negrin have edited the manuscript for English language. They are native English speakers and have worked on phytochemical studies for many years. Some corrections have been made.
Reviewer 2 Report
Y Ji et al in the publication “Cytotoxic xanthones from Hypericum stellatum, an ethnomedicine in southwest China” described isolation from Hypericum stellatum species, determination of structure and assessment of cytotoxic activity in liver cancer cell lines and one normal cell line. The two new compounds were identified. The ability of newly and previously isolated compounds to inhibit cell growth was studied with MTS method and reviled that the anticancer activity of new compounds was rather moderate, however the prenyl group was identified as a beneficial with respect to molecule cytotoxicity. The novelty of presented paper is moderate. Moreover there is a number of issues which should be addressed:
What was the reason for studying the anticancer activity of compounds, please provide the rationale. According to conclusion the authors declare that this research found hepatic protection compounds from H. stellatum. Was the aim of the study to find protective compounds? The study in normal cell line showed that compounds exhibited rather high toxicity toward normal cell line (a model of healthy hepatic tissue?). On the other hand the results in cancer cell line indicate some toxicity what could suggest anticancer mode of action. With respect to this the conclusion is not clear and not adequate to the results obtained. There is a lack of molecular description of cell line studied – is this the same type of cancer, why these lines were chosen, please justify. The strongest activity was exhibited by known compound Isojacareubin. The anticancer activity of Isojacareubin was previously shown in hepatic cancer cell lines. The authors should at least refer to this publication (Yuan X, et al, Sci Rep. 2015;5:12889) What was a rationale for addition of 4 additional xanthones? The strongest activity was exhibited by known compound Isojacareubin, however the authors noticed that the compound was also cytotoxic to ward normal cells. The other compounds selectivity should be discussed MTS assay description: Taxol or paclitaxel? Please add the name of cisplatin provider please verify the declared number of compounds identified and used in the cytotoxicity assay in different parts of publication, it seems there is some differences Please place the number specifying each compound when first mentioned, and also place the name Isojacareubin when first time mentioned. Lines 127 – 137: The whole paragraph is hard to read, moreover in lines 135-136 and in line 134 different compounds (2,7,8 or 1, 4, 9 and 10) were mentioned to have moderate activity in HepG2 cells in two separate sentences. It would be beneficial if the author list the activity of the compounds in more logical way. line 139: 9 shoul be bold line 161, 246: please provide reference
Author Response
Thank you very much for your comments. We respond to these questions in the following points:
1. What was the reason for studying the anticancer activity of compounds?
In our field investigations, we obtained information that Hypericum stellatum has been used to treat hepatitis and hepatocellular carcinoma. So we hypothesized that Hypericum stellatum contains compounds that are effective in the liver.
Hepatocellular carcinoma (HCC) is the end-stage clinical outcomes of chronic hepatitis B (HBV) and chronic hepatitis C (HCV). The prevalence of HBV and HCV infection are high in the East, Southeast and Central Asia and the sub-Saharan Africa. HBV is more prevalent in patients with HCC in China.
In our research, we isolated ten xanthones from Hypericum stellatum. A review of the research literature indicates that xanthones are potent cytotoxic compounds, active against various cancer cells. Based on data from our fieldwork, we chose to target human liver carcinoma cell lines for further study.
2. Was the aim of the study to find protective compounds?
The research was inspired by the traditional medicinal usage of H. stellatum. Our study aimed to explore the phytochemistry and bioactivity of H. stellatum to explain why local people widely applied H. stellatum for liver protection, to treat hepatitis, and to fight against liver cancer. Based on this rationale, we conducted our investigation and identified several cytotoxic xanthones.
3. The study in normal cell line showed that compounds exhibited rather high toxicity toward normal cell line.
We evaluated the xanthones from two aspects: Cytotoxicity toward liver carcinoma cell lines and toward normal liver cell lines. High cytotoxicity toward liver carcinoma cell lines and low cytotoxicity toward normal liver cell lines indicated that the compounds have potential to be considered as active medicinal compounds, warranting further investigation. If the compound had both high cytotoxicity toward liver carcinoma cell lines and normal liver cell lines, then structural modification would be recommended.
4. The results in cancer cell line indicate some toxicity what could suggest anticancer mode of action.
Some compounds showed cytotoxicity towards five human liver carcinoma cell lines. As cytotoxicity is not equivalent to anticancer activity, we presented the results for the cytotoxicity of xanthones.
5. There is a lack of molecular description of cell line studied – is this the same type of cancer, why these lines were chosen, please justify.
Five human liver cancer carcinoma lines were used which possess different biological characteristics:
SMMC-7721: It was derived from a liver hepatocellular carcinoma of a 50 year old Chinese male, AFP immunofluorescence reaction is positive.
Huh-7: It was derived from a liver hepatocellular carcinoma of a 57 year old Japanese male, at a level of growth, morphology, and HCVcc permissiveness.
HepG2: It was derived from a liver hepatocellular carcinoma of a 15 year old Caucasian male. The cell line is well suited for cytogenetic studies, as it retains at least some biosynthetic and metabolic capabilities characteristic of normal human liver parenchymal cell.
SK-HEP-1: Human liver adenocarcinoma cells which were derived from adenocarcinoma of a 52 years old Caucasian male; the SK-HEP-1 line has been identified as being of endothelial origin.
PLC/PRF/5: It was derived from hepatoma. The line was originally contaminated with mycoplasma, and was cured by treatment with BM-cycline. The cells secrete HBsAg.
These five human liver carcinoma cell lines are representative of a broad sample of human liver cancer types, so we choose these five cell lines to evaluate the effect of xanthones on human liver carcinoma cell lines.
6. The strongest activity was exhibited by known compound Isojacareubin. The anticancer activity of Isojacareubin was previously shown in hepatic cancer cell lines. The authors should at least refer to this publication (Yuan X, et al, Sci Rep. 2015;5:12889)
Thank you very much! We refer to this paper and have cited it in the revised version.
7. What was a rationale for addition of 4 additional xanthones?
We isolated the 4 xanthones before in the same plant extracts, together with 6 xanthones isolated in this research. Using these 10 xanthones with different substituent groups we conducted the cytotoxicity assay to analyze the bioactivity and assess structure-activity relationships.
8. The strongest activity was exhibited by known compound Isojacareubin, however the authors noticed that the compound was also cytotoxic toward normal cells. The other compounds selectivity should be discussed.
We evaluated the xanthones from two aspects: The cytotoxicity toward liver carcinoma cell lines and the cytotoxicity toward normal liver cell lines. We discussed details in Lines 132-143.
We have added the name of the cisplatin provider, and we adjusted the number of compounds to make it clear. Lines 127-137 (After revision, these are now lines 132-143) were reorganized to make it more clear and logical. Other corrections have been done according to your kind comments.
Reviewer 3 Report
the article is pleasant to reading and as far as my English level is acceptable appears to be of good quality. However however a sentence in the conclusion is really not acceptable and not supported by the biological studies. In vitro cytotoxic studies cannot allow any conclusion about an hepatic protection during human usage. In order to be published the sentence "Results provide scientific evidences that there are hepatic protection compounds in H. stellatum..." must be replaced by another sentence explaining that complementary experiments must be done to try to explain an hepatic protection. Another way of doing things may be to explain how it will be possible to demonstrate the hepatic protection by in vivo experiment with purified compounds in animal models of hepatic dysfunction.
Author Response
Thank you for your comments. We have replaced the sentence with the following: “complementary experiments must be done to clarify the biological mechanism of H. stellatum hepatic protection”. This is also the focus of our research group in the future.
Reviewer 4 Report
This contribution reports the isolation, identification and cytotoxicity evaluation of xanthones from the endemic Chinese plant Hypericum stellatum. Two new compounds were identified and their structures appropriately characterized. Some isolated xanthones showed moderate and strong cytotoxic effects, one of them with IC50 values in the single-digit micromolar concentration range against 4 cell lines, being lower hat those of the standard anticancer agent cisplatin. I consider his manuscript to be well written and organized. The introduction, discussion part and experimental details are appropriate. In overall, I think this manuscript is of good quality, with results that are interesting in the fields of phytochemistry and medicinal chemistry, contributing to the knowledge on the biological effects of Hypericum extracts with the disclosure of promising structures of anticancer potential. I do not have any corrections or improvements to suggest. Therefore, I recommend its publication in the present form.
Author Response
Thank you so much!
Round 2
Reviewer 2 Report
Dear Authors, I think that the authors must have sent the wrong version of corrected manuscript. In the manuscript text, only English editing is marked and no other changes have been done, although the authors stated in the Author's respond letter that they have introduced changes into the manuscript.The authors answers are satisfactory but the text should be changed accordingly.
Hence I am not able to assess the corrected manuscript.
Author Response
Thank you so much for your comments. Yes, you are correct! I uploaded a wrong version of our revised manuscript on September 24, which was not the latest version. I apologize. We now repeat point-by-point response as follows:
1. What was the reason for studying the anticancer activity of compounds?
In our field investigations, we obtained information that Hypericum stellatum has been used to treat hepatitis and hepatocellular carcinoma. So we hypothesized that Hypericum stellatum contains compounds that are effective in the liver.
Hepatocellular carcinoma (HCC) is the end-stage clinical outcomes of chronic hepatitis B (HBV) and chronic hepatitis C (HCV). The prevalence of HBV and HCV infection are high in the East, Southeast and Central Asia and the sub-Saharan Africa. HBV is more prevalent in patients with HCC in China.
In our research, we isolated ten xanthones from Hypericum stellatum. A review of the research literature indicates that xanthones are potent cytotoxic compounds, active against various cancer cells. Based on data from our fieldwork, we chose to target human liver carcinoma cell lines for further study.
2. Was the aim of the study to find protective compounds?
The research was inspired by the traditional medicinal usage of H. stellatum. Our study aimed to explore the phytochemistry and bioactivity of H. stellatum to explain why local people widely applied H. stellatum for liver protection, to treat hepatitis, and to fight against liver cancer. Based on this rationale, we conducted our investigation and identified several cytotoxic xanthones.
3. The study in normal cell line showed that compounds exhibited rather high toxicity toward normal cell line.
We evaluated the xanthones from two aspects: Cytotoxicity toward liver carcinoma cell lines and toward normal liver cell lines. High cytotoxicity toward liver carcinoma cell lines and low cytotoxicity toward normal liver cell lines indicated that the compounds have potential to be considered as active medicinal compounds, warranting further investigation. If the compound had both high cytotoxicity toward liver carcinoma cell lines and normal liver cell lines, then structural modification would be recommended.
4. The results in cancer cell line indicate some toxicity what could suggest anticancer mode of action.
Some compounds showed cytotoxicity towards five human liver carcinoma cell lines. As cytotoxicity is not equivalent to anticancer activity, we presented the results for the cytotoxicity of xanthones.
5. There is a lack of molecular description of cell line studied – is this the same type of cancer, why these lines were chosen, please justify.
Five human liver cancer carcinoma lines were used which possess different biological characteristics:
SMMC-7721: It was derived from a liver hepatocellular carcinoma of a 50 year old Chinese male, AFP immunofluorescence reaction is positive.
Huh-7: It was derived from a liver hepatocellular carcinoma of a 57 year old Japanese male, at a level of growth, morphology, and HCVcc permissiveness.
HepG2: It was derived from a liver hepatocellular carcinoma of a 15 year old Caucasian male. The cell line is well suited for cytogenetic studies, as it retains at least some biosynthetic and metabolic capabilities characteristic of normal human liver parenchymal cell.
SK-HEP-1: Human liver adenocarcinoma cells which were derived from adenocarcinoma of a 52 years old Caucasian male; the SK-HEP-1 line has been identified as being of endothelial origin.
PLC/PRF/5: It was derived from hepatoma. The line was originally contaminated with mycoplasma, and was cured by treatment with BM-cycline. The cells secrete HBsAg.
These five human liver carcinoma cell lines are representative of a broad sample of human liver cancer types, so we choose these five cell lines to evaluate the effect of xanthones on human liver carcinoma cell lines.
6. The strongest activity was exhibited by known compound Isojacareubin. The anticancer activity of Isojacareubin was previously shown in hepatic cancer cell lines. The authors should at least refer to this publication (Yuan X, et al, Sci Rep. 2015;5:12889)
Thank you very much! We refer to this paper and have cited it in the revised version.
7. What was a rationale for addition of 4 additional xanthones?
We isolated the 4 xanthones before in the same plant extracts, together with 6 xanthones isolated in this research. Using these 10 xanthones with different substituent groups we conducted the cytotoxicity assay to analyze the bioactivity and assess structure-activity relationships.
8. The strongest activity was exhibited by known compound Isojacareubin, however the authors noticed that the compound was also cytotoxic toward normal cells. The other compounds selectivity should be discussed.
We evaluated the xanthones from two aspects: The cytotoxicity toward liver carcinoma cell lines and the cytotoxicity toward normal liver cell lines. We discussed details in Lines 132-143.
We have added the name of the cisplatin provider, and we adjusted the number of compounds to make it clear. Lines 127-137 (After revision, these are now lines 132-143) were reorganized to make it more clear and logical. Other corrections have been done according to your kind comments.
Reviewer 3 Report
The article is interesting but I'm sorry to say that the following sentences in the abstract and conclusion are not acceptable!!
in the abstract : "This research found hepatic protection compounds from H. stellatum". cannot be accepted, in vitro cytotoxicity studies cannot allow any conclusion about an in vivo hepatoprotection???!!!
In the conclusion
"These results provide scientific evidences that there are evidence for the presence of hepatic protective compounds in H. stellatum, " sentence not acceptable!! I completely agree with the fact that further experiment will try to demonstrate the mechanism of hepatic protection but current results demonstrate that some components are cytotoxic which is not a good argument to demonstrate that an hepatoprotection exists...
Author Response
Thank you very much for your comments. I wrongly uploaded a version of our revised manuscript which was not the latest version. I aapologize.
In the latest version, they were changed.
Thank you again!